

# Optimum Coagulant Forecasting with Modeling the Jar Test Experiments Using ANN

Sadaf haghiri[1], Sina Moharramzadeh[2], Amin Daghighi[3,4]

[1]Department of Environmental Engineering Faculty of Middle East Technical University, Turkey

[2]Department of Civil, Construction and Environmental Engineering, Iowa State University, Iowa, USA

[3]Department of Civil and Environmental Engineering, Cleveland State University, Ohio, USA

[4]Consultant Engineer at Daneshkar Ahwaz Company, Tehran, Iran

*Correspondence to*: Amin Daghighi (a.daghighi@vikes.csuohio.edu)

**Abstract.** Nowadays the proper utilization of water treatment plants and optimizing their use is of particular importance. Coagulation and flocculation in water treatment are among the common ways through which the use of coagulants leads to instability of particles and formation of larger and heavier particles, resulting in improvement of sedimentation and filtration processes. Determination of the optimum dose of such a coagulant is of particular significance. A high dose, in addition to

adding costs, can cause the sediment to remain in the filtrate, a dangerous condition according to the standards, while a sub-adequate dose of coagulants can result in the reducing the required quality and acceptable performance of the coagulation process. While jar tests are used for testing coagulants, such experiments face many constraints with respect to evaluating the results produced by sudden changes in input water because of their significant costs, long time requirements, and complex relationships among the many factors (turbidity, temperature, pH, alkalinity, etc.) that can influence the efficiency of coagulant

and test results. Modeling can be used to overcome these limitations, and in this research study, Artificial Neural Network (ANN) Multi-Layer Perceptron (MLP) with one hidden layer has been used for modeling the jar test to determine the dosage level of used coagulant in water treatment processes. The data contained in this research have been obtained from the drinking water treatment plant located in the Ardabil province. To evaluate the performance of the model, the parameters Mean Squared Error (MSE) and the Correlation Coefficient $R^2$ have been used. The obtained values are within an acceptable range that

demonstrates the high accuracy of the models with respect to the estimation of water quality characteristics and the optimal dosages of coagulants, so using these models will allow operators to not only reduce costs and time taken to perform experimental jar tests, but also to predict a proper dosage for coagulant amounts and to project the quality of the output water under real variable conditions.

**Key words:** Modeling, Artificial Neural Networks, Water Treatment, Testing, Current Testing.



## 1 Introduction

When we speak of water refinery operations, we usually mean that the final price for the produced water must be decreased in a way that achieves an optimum combination of efficiency and affectivity. The main goal of this research study is the management of chemical substances to decrease the final cost of water in which, for very similar inputs, the amount of

coagulating chemicals required to decrease water turbidity is determined.

Water turbidity is one important and significant parameter when water refineries obtain their input water from natural resources like rivers or lakes. In a water refinery, water turbidity must be resolved along with water sterilization to ensure water clarity; otherwise the refined water is in no way usable for drinking purposes. Materials called coagulators are used to decrease water turbidity, with the amount of required coagulator different depending on the environmental conditions like temperature, pH

value, the amount and the type of turbidity, etc., and required amount is usually determined by performing an experiment called a jar test. Since these experiments are time-consuming and also contain errors, they do not always provide a correct estimate of the optimum amount of coagulating substance and they may increase the cost of the chemicals used and diminish management capability of the refineries to appropriately decrease the chemical requirements. Threfore, proposing new models for estimating the optimum amount of coagulating materials the experimental results seems to represent an appropriate way to

alleviate costs and the time requirements and generally improving the health conditions of drinking water (Lamrini et al., 2005).

Most particles that cause water to become opaque have the feature of hydrophobicity and are often settled by simple gravitational force as time passes, but there may be some smaller particles that cause water color, like Hydro oxides known as colloidal or acids such as Humic acid and Folic acid that are organic acids. These are all hydrophilic and do not tend to be

settled. Bacteria are also colloidal particles and are not separated in the basic coagulation phase (Franceschi et al., 2002).

Modeling is an important math-based tool typically performed in one of the two following ways:

- Numeric or Deterministic Methods

- Statistical Methods

Statistical methods are divided into classical (multi-variable regression) and advanced methods. While conventional modeling

has been used to describe biological procedures has been based on writing the equations for the velocity of the growth of the microorganisms, the consumption of the substrate, and the forming of the product, because microbiological reactions are no-linear and time-dependent with a rather complicated nature, such modelings have many restrictions. The structure of statistical models is often simpler than that of deterministic models and they provide a more general view of the nature of the issue that recognize the applicable relationships between the efficient parameters as a necessary part of the problem. Also, in gaining

knowledge of the relationships between the parts of the model, a need arises for solving complicated equations, and in some situations the answers cannot be obtained under general conditions. Numerical methods will thus not be used in this study. In the linear multiple-variable regression model, there are a great many assumptions, and using them all in practical problems causes problems. In performing research, because these assumptions represent complex statistical issues, they many be hidden



from researchers' eyes and not taken seriously (Daghighi, 2017), so the model proposed woul;d not have the required accuracy, so these types of models cannot be used in problems demanding high accuracy (Homada and Al-Ghusian, 1999).

Artificial neural networks (ANNs) are one of the advanced statistical methods used in many the scientific fields today. Using a neural network an advanced statistical method able to anticipate non-linear and complex relationships between inputs and outputs, and is often used to replace linear multiple-variable regression. ANNs are a set of non-linear techniques that do not require choice of a pre-determined mathematical model, because the relationships between input and output variables are automatically set by the utilized algorithm, so neural networks can be a proper choice for solving those problems in which certain relationships between the variables are either not known or describing them would be difficult. The neural network was first proposed by McCulloch and Pitts in 1943.

The regular architecture of ANNs consists of three layers: the input layer (distributes the data sent into the network), the hidden layer (processes the data) and the output layer (extracts the results per certain inputs). A network can have several hidden layers.

Theoretical tasks performed in the field have shown that a hidden layer for these models can approximate virtually any complicated and non-linear function (Maier and Dandy, 2005), as proven by experimental and practical results (Homada and Al-Ghusian, 1999).

Categorizing variables for using them in a neural network can be performed in many different ways. In this research, the parallel validity recognizing method, a method often used in the neural networks models, is used. for determining the time of the end of the education and comparing the generalization capability of different models. In the parallel validity recognizing method, a separate group of data is used for experimentally examining the ability of different models to generalize with respect with the different levels of the education. Since data for this separate group must not be in the two groups of education and validity recognizing, in this method the data are randomly categorized into three groups: the group related to the network education (about 80 percent of all the input data), through which network weights are determined: the group monitoring network education (about 10 percent of all the data), through which the network error is studied to reeducate the network with respect to stopping calculations to make a decision: and the validity-recognizing group (about 10 percent of the data) that studies the capability of the network after its education. Education occurs up to the time atwhich the error related to the monitoring data group decreases, at which point the education is stopped. Using this method, also called the Stop Training Algorithm, use of more complicated architectures in designing the network is provided to the operator without having any over-fitting problem, and by using some of the factors happening while the problem is in the network, the education can stopped. The standards mentioned play an important role in this method (Standard Methods, 1998).

## - The Performed Experimental Researches

The ANN technique was first proposed by McCulloch and Pitts in 1943. Despite using a very simple structure, its velocity and the power of the calculation was strongly noted. The typical architecture of an ANN consists of three layers: the input layer (distributes the data in the network), the hidden layer (processes the data) and the output layer (extracts the results per certain

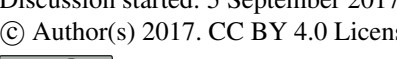



inputs). A network may have several hidden layers. The theoretical tasks that have been performed this field show that a hidden layer can approximate many complex and non-linear functions (Cybenko, 1989; Hornik et al., 1989), as confirmed by experimental and practical results (Oliveira-Esquerre et al., 2002). Categorizing variables for use them in a neural network can be performed using different methods, and in this study the parallel validity recognizing method a method previously used

several times in neural networks models, is used. This method is used for determining the education end time and comparing the capability for generalizing different models (Bowden et al., 2002). In the parallel validity recognizing method, a separate group of data is used for experimenting with the power of generalizing the model with respect to different levels of education. Use of a separate group means that the data must not be in the either the education or validity-recognizing groups, and the data are randomly categorized into three groups: The group related to the network education (about 80 percent of all the input data)

in which the network weights are determined; the group monitoring the network education (about 10 percent of all the data), used to study the network error of this while reeducating the network to discover when calculations should stop and the decision made; and the validity-recognizing group (about 10 percent of the data) that studies the network capability after education. Education occurs up to the time at which the error related to the monitoring data group decreases, at which point the education is stopped. Using this method, also called the Stop Training Algorithm, capability for using more complicated

architectures in designing the network is provided for the operator without having any over-fitting problem, and by using some of the factors discovered while the problem remains in the network, the education is stopped. The standards mentioned therefore play an important role in this method (Standard Methods, 1998).

Recent research studies show that determining statistical dimensions can enhance the capability and improve the performance of neural models (Maier H. R. and Dandy G. C., 2000). Analyzing the main components is a technique for transforming the

orthogonal components and if needed, for decreasing the number of dimensions of the variables, decreasing the number of variable dimensions used for correction, and improving the operation of the models developed in the neural network. This technique is a regular and practical method for data with several dimensions (Bui et al., 2016). The issue of recognizing the pattern which the data obeys–especially when there are more than two dimensions- is a very difficult work and the relationships usually cannot be graphically depicted, so this method is for analyzing the problems in which several factors are important in

the issue studied. The use of this technique smooths the path by making a model for simulating the biological refines process using a neural network. Research studies have shown that the simultaneous application of the technique of combined analysis of the main components and the ANN produces better and more accurate results than the situation of considerably them separately (Oliveira-Esquerre et al., 2002), so this study will use the same method (Zhang and Stanley, 1999; Baxter et al., 1999) by using the Multi-Layer Perceptron (MLP) structure for the ANN and its modeling for anticipating the turbidity and

the color of the refined water at the Rosedale Refinery in Alberta, Canada. Gagnon (Gagnon et al., 1997) using the method of inverse models in neural networks, a similar method, for anticipating the necessary amount of Alum for the Ste-Foy Refinery in Quebec, Canada (Joo et al., 2000) A modeling has also been performed for the Chungju Refinery in Korea. Van Leeuwen prepared a neural network model based on the jar test procedure for use on the collected surface waters in southern Australia



(Van Leeuwen et al., 1999). In the studies mentioned, because of the lack of proper output parameters, the ruling equations of the prepared models were not able to adapt themselves to output variable changes.

Zhang and Stanley (1999) added the refined water turbidity factor as an input parameter to the water characteristics parameters in their neural network for anticipating the optimum amount of Alum for use in the Rossdale Refinery. Yu did the same for
the Taipe Refinery in Taiwan by applying a greater number of parameters, preparing his 3 neural network models for anticipating the proper amount of Alum necessary for coagulation (Yu et al., 2000). According to the studied background, this method can be used for anticipating the proper amount of coagulating material, that in this research will be determined using the data from the Ardebil province drinking water refinery and determining the available effective factors in the neural network, including the error percentage that can be passed up, the amount of experimental expenses, and the time needed for performing
the jar test, all of which can be reduced by the preparing the model and using the results.

Modeling by the use of neural networks reduces test time and cost, and decreases the necessity of performing experiments now being performed in the drinking water refinery of Ardebil province. In addition, in this method there is no need to know the type of the input and output parameters and the quality of performing the process, and by only providing the data to the program in numeric form, the determined answer is obtained along with all the effective factors on the process (which in this project
are the temperature, pH, the degree of alkalinity, and the turbidity, according to the accessible data of the refinery) hidden in the data. While analyzing the theory of the process of accessing to a mathematical and analytical relation, there is a need to determine the assumptions related to systems simplification, possibly eliminating some of the effective factors. This issue proves the capability of this type of neural network in recognizing complex and unknown systems. Thus, in this research, the following goals are being studied: Anticipating the optimum amount of the coagulating material, Analyzing the collected data,
Determining the best type of the neural network for the purpose of modeling with the lowest error, Determining the optimum amount of coagulating material in the process of coagulation and flocculation, Validity recognition of the operation of the developed models.

Using this model would allow the operators of a water refinery to avoid time and cost wasteage, and decrease the need to perform some of the experiment are now being performed in the drinking water refinery of Ardebil.

**2 Methodology**

**2.1 Data**

The available data related to the drinking water refinery in Ardebil province have been collected over two years and the change ranges for them are presented in Table 1.

**Table 1: Ranges of available data**

| | The characteristics of the input water | | | | The characteristics of the output water | | | |
|---|---|---|---|---|---|---|---|---|
| | pH | Temperature | Alkalinity | Turbidity | Alkalinity | Temperature | pH | The Final Turbidity |
| MAX | 7.9 | 24 | 264 | 16 | 235 | 17.5 | 8.2 | 1.3 |
| MIN | 7.1 | 7 | 201 | 7 | 180 | 9.5 | 7.2 | 0.5 |



**2.2 Data Division and Pre-Processing**

The 112 accessible information points have been categorized into three groups using of the Bowden model (Bowden et al., 2002). The groups are: 1. training groups for setting up the connection weighs, 2. A testing group for knowing when to cease

training and optimizing the structure of the neural network and the specifications of the internal model (for example,  the rate of learning, the momentum) and 3. A validating group for testing the model's capability for generalizing the model for the range of the information used for calibration. This method utilizes a Self Organizing Map (SOM) for categorizing the high dimensional input-output information in two-dimensional space (Kohonen , 1982). Then information to be used for training, testing and the validation have been chosen and, as a result, they contain values from each group. This act makes sure that all

three groups of information have all the data patterns and so properly represent the statistics of the population. Clustering the information into the groups of testing and training makes sure that over-fitting will not happen and that the data used for validation will not be not utilized for developing the model in any capacity.

The use of this method has resulted in 80, 10, and 10 percentage information points in the training, testing, and validation, respectively.

In both prepared models, all data were normalized before being entered into the model using the normal distribution function:

$$z = \frac{(x - \bar{x})}{S} \tag{1}$$

In which, $x$ is the primary amount, $\bar{x}$ is the average of the data, and $S$ is the standard deviation.

**2.3 Choice of Model Inputs**

**2.3.1 Prediction of treated water quality parameters**

The model for anticipating the quality of the output water was first prepared. In this model, after normalizing the input data, an MLP ANN with a hidden layer and 15 neurons (the result of the trial and errors) was used. In this model, the data related to the pure water (the turbidity, the alkalinity the temperature, and the pH) and the amount of coagulating materials the model input and the quality of the output water (turbidity, temperature, the amount of being alkaline and pH) have been included The Figure 1 (a) to (d) are related to the data validations that have been prepared by calculating the error of the model and analyzing

the amount of difference between the real data and the anticipated ones reached using 10 percent of the data.

**2.3.2 Prediction of Optimal Dose (Process Inverse Model)**

Model 2, representing the goal of this research, anticipates the amount of necessary coagulating material according to the input characteristics of the input water and the desired standard water characteristics. In this model, after normalizing the input data, an MLP ANN with a hidden layer and 16 neurons produced by trial and errors) was used. In this part, according to the real

inputs taken from the water refinery, the final model was built, and the necessary pure water and the output water characteristics

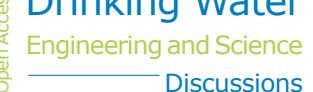



are given to the model, with the amount of the necessary coagulating material is given as the output. Table 2 demonstrate the inputs of the both model.

**Table 2: Models inputs and outputs**

|  | The characteristics of the input water | | | | The characteristics of the output water | | | | Alum dose |
|---|---|---|---|---|---|---|---|---|---|
|  | pH | Temperature | Alkalinity | Turbidity | pH | Temperature | Alkalinity | The Final Turbidity |  |
| Mod 1 | I | I | I | I | O | O | O | O | I |
| Mod 2 | I | I | I | I | I | I | I | I | O |

## 3 Results and Discussions

5   The results produced by using model 1, related to the pH, alkalinity, temperature, and turbidity, are shown in Figure 1 (a-d), with the related amounts of $R^2$ and Mean Squared Error (MSE) shown in Table 3. The model has accomplished accuracy in predicting all the three parameters of the treated water quality, the $R^2$ values from 0.94 for anticipating the amount of alkalinity and 0.85 for anticipating the pH. The mean square errors for the prediction of turbidity are also properly small (errors of 0.011 NTU, 0.01 pH, 0.67 °C and 26.31 mg/lit).

**Table 3: The Results of the Neural Model No.1**

| Model | Output | $R^2$ | MSE |
|---|---|---|---|
| 1 | pH of Treated water | 0.85 | 0.01 |
| 1 | Alk of Treated water | 0.94 | 26.3 |
| 1 | Turb of Treated water | 0.9 | 0.024 |
| 1 | Tempt of Treated water | 0.93 | 0.67 |





**Figure 1: Scatterplots of actual versus predicted values for the validation data obtained using model 1 (a) Actual pH, (b) Actual Alkalinity, (c) Actual Turbidity, and (d) Actual Tempt**

The results related to the residual aluminum and pH obtained by using model 2 are shown in Figures 2 and 3, with the corresponding amounts of $R^2$ and MSE shown in Table 4. The model has also achieved accuracy s in predicting the amount of the residual aluminum of the treated water with an $R^2$ of 0.93 and a MSE of 0.37 mg/l.



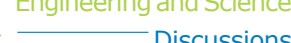
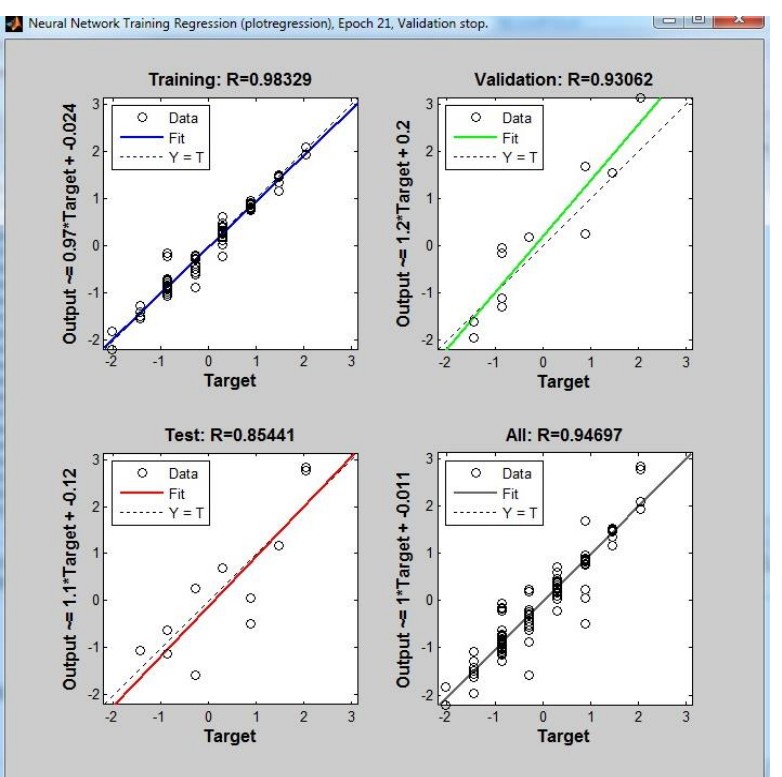

**Figure 2: Comparing the real data with predicted data in 3 steps and in form of general**

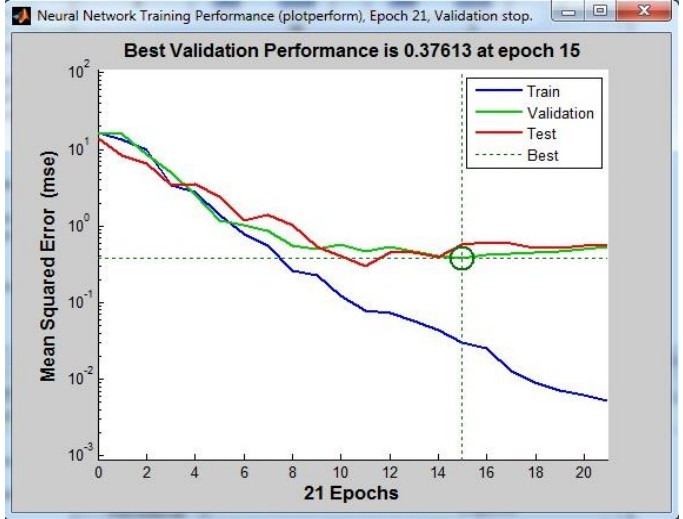

**Figure 3: Related figure for learning model No.2 and validation of model**

**Table 4: The Results of the Neural Network No.2**

| Model | Output | $R^2$ | MSE |
|---|---|---|---|
| 2 | Coagulant dose | 0.93 | 0.37 |



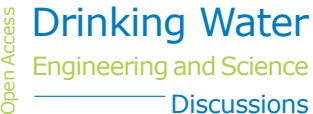

Finally, a user-friendly model has been prepared for use by the operator. The prepared software is a Graphical User Interface (GUI) generated by MATLAB software. Figure 4 illustrates how the operator can observe the results related to output water quality by entering the input water data and the amount of coagulating material in the first model.

5   Also for model No.2 another user-friendly model was programmed containing various parameters such as ph, alkalinity, temperature, and turbidity for both raw and treated water. After the model has been the recommended dosage of coagulant (in this research Alum is the coagulant material) is achieved which are shown in Figure 5.

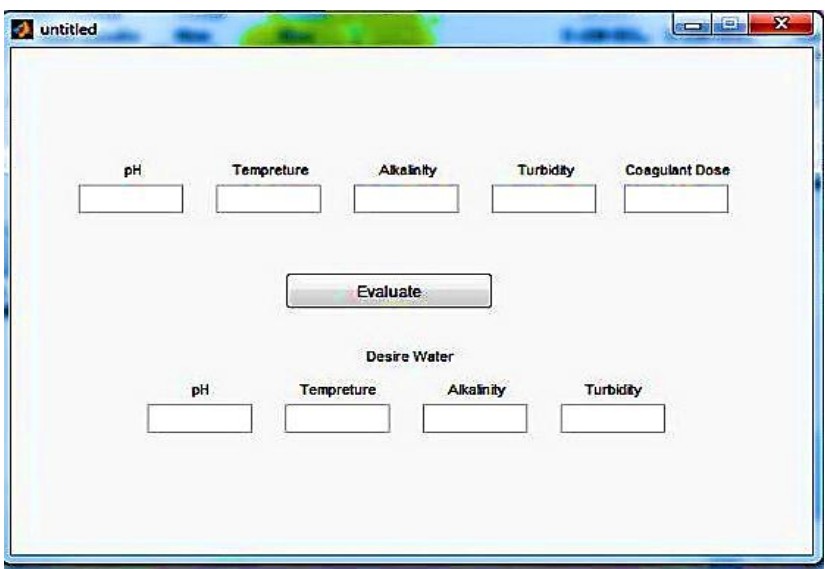

10   **Figure 4: The Graphical User Interface (GUI) of the Model No.1**

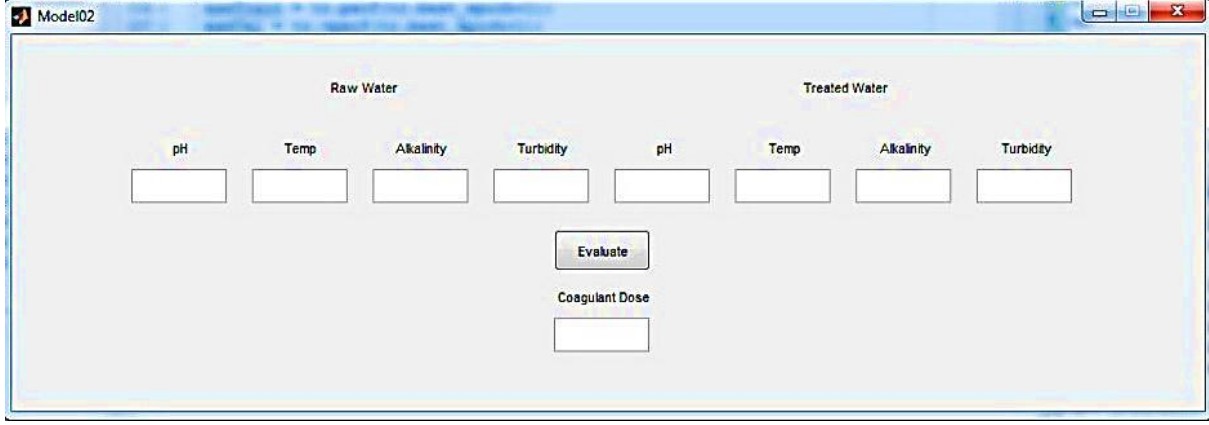

**Figure 5: The Graphical User Interface (GUI) of the Model No.2**





## 4 Conclusions

In this research, the simulation of the jar experiment in the flocculation and coagulation unit at the Ardebil province drinking water refinery have been studied using for the first time a neural network of type MLP. Using this method, two models were created to enable presentation of water quality characteristics after coagulation and flocculation and anticipating the optimum amount of the coagulating material related to changing characteristics of the input water in the minimum possible time and by the highest accuracy.

In choosing effective parameters in the jar test, some parameters such as the color and hydraulic parameters like the clarifier overflow rate, etc., have been taken into consideration, in addition to the parameters considered in simulating this research (temperature, pH, turbidity, and the amount of coagulating material). As in previous studies, because of not measuring the number of parameters listed and not recording related data, and also because of the relatively small effect they have in comparison to the considered parameters, we were not able to apply their effects. Of course the considered parameters are among the most important effective parameters, and because extending all the effective elements in a single experiment was not feasible, it is possible to consider the prepared model for the results of this experiment as a good approximation.

Because the neural network model is a parametric method, if the amount of recorded data is increased, the model accuracy will increase. In this study, there were 112 recorded models used over a two-year period, that included data related to the input (pure) water, the output water, and the proper amount of \ coagulating material. Because of the absence of high rate changes among the data and the absence of outline data, there was no need to prepare, analyze, and select the data. Since the utilized components have different units and dimensions, they cannot be compared in terms of average and standard deviation, so they have been normalized. The training of the model has been performed well according to analysis of existing errors, so the most prominent and the only weak point is access to only a small amount of recorded data.

One notable suggestion for continuation of the study would be utilization of other modeling methods, especially other types of neural networks, and also the utilization of the resources with a greater amount of recorded data related to the jar experiment.

## 5 Acknowledgments

With the great thanks to Professor Naser Mehrdadi and Water Institute of University of Tehran.

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

)