# Peer review of "Optimum Coagulant Forecasting with Modeling the Jar Test Experiments Using ANN"

_Drinking Water Engineering and Science, 2017_

## Referee Comment (RC1) · S.R Mounce (Referee) · 25 Sep 2017

General comments–

This paper presents application of MLP ANNs to optimum coagulant forecasting for water treatment. Two models are developed, the principle one being the prediction of optimal dose. R2 of 0.93 was achieved for this model. Four variables were used. A two year historical dataset was utilised (112 data points). A GUI was developed of the deployed model for operators.

This is a useful real world application of neural network technology, however the paper could be significantly improved and clarifications made.

The authors did no discuss the limitations of the data set they used, i.e. only 112 data

point. In sections 2.2 and 2.3 little actual implementation details were provided. Such as parameter settings of the ANN model. They should reference what ANN software they used. Was it MATLAB ANN toolbox or other? What did they use for the SOM they used for splitting the data? Also, training, validation and testing results are not clearly provided.

'-The Performed Experimental Researches' is a mishmash of lit review material that has already appeared, domain specific review and case study info. It needs restructuring and repetition removing, see below.

The authors do not mention latest developments in ANNs such as deep learning.

The conclusions need work, see below.

Specific comments and technical corrections–

p1 line 11. Suggest don't start an abstract with 'Nowadays'- 'Currently'?

p1 line 23 Which country?

p1 line 24-25 vague. Include some actual figures on accuracy/ performance

p2 First 2 paragraphs. Lack of references to set the scene

p2. Only deterministic or statistical? Data driven/ machine learning (such as ANNs) is not really a subset of statistical (you describe this as 'advanced statistical'). Different fields. I'd revise this into:

Deterministic Data driven. Subsets: classical statistical. Machine learning.

p2. line 23-26. Talk about data driven?

p3 line 1 'would;d'

p3 first paragraph change 'statistical' to 'artificial intelligence' or 'machine learning'

p3 Line 9 ref?

p3 line 9. Isn't the Hornik ref more appropriate? USe the Maier & Dandy ref for talking about ANNs applied to water resources.

p3 line 17 ref for method.

p3 line 17 Remember you can use your ANN acronymn since you defined it.

p3 line 18, 20, next 2 paragraphs. Education?! You mean 'training'

p3 lines 20-30. More commonly referred to as training, validating and testing in ANN usage. And actually, you use this on page 6 anyway

p3 Line 30. '-The Performed Experimental Researches' What is this section? A complete mix of the previous introduction, often repeating (quoting word for word) what we have already read, background on literature review and information about the case study. Very confusing, this needs correcting.

Have a new 'Background section' and finish with a paragraph containing the case study info?

p3 Lines 31-33 Repetition from introduction

p4 Lines 1-17 Repetition from introduction - but with different references!!

p4 Line 18 - p5 line 25 This should now be in 'Background'.

p4 line 27 change 'considerably' to 'consideration of'

p4 line 32 change 'A modeling' to 'Modelling'

p5 lines 11-24. Is this better in a 'case study' section? Perhaps an extra section in 2.

p5 line 16-17 - opaque sentence...

p5 line 24 change 'experiment now' to 'experiments that are now'

p5 Table 1. You should specify the units of measurement

p6 Lines 7 to 12. Do you have high dimensional data here? There are 8 dimensions in table 1

p6 line 21 Specify how many input and output neurons

p6 lines 22-23. Confusing. Should it be 'was the model input'? Also, a full stop is missing. Also, what is the target output exactly?

p6 line 29 Specify how many input and output neurons

p6 line 29 Missing bracket '(produced by trial and error)'

p7 line 2 change 'of the both model' to 'of both models'

p7 line 5 - already introduced

p7 lines 6 to 8. Is R2 0.85 'accuracy'? Change to 'reasonable accuracy'

p7 line 8 change 'properly' to 'relatively'

p8 line 7 change 'accuracy s' to 'reasonable accuracy'

p8 line 6. Why is PH mentioned. Isn't residual aluminium the output?

p9 Figure 2. Make it clear what the variable is i.e. of real/predicted (I assume residual Almunium)

p9. Table 4. Should provide train/ test results.

p10. Please correct the sentence "After the model has been the recommended dosage of coagulant (in this research Alum is the coagulant material) is achieved which are shown in Figure 5."

p10. Authors should be commended for providing a practical tool. You only need one screenshot though or it can start to look like a manual. I suggest you go with Fig 4 and not Fig 5. Could this be in a discussion & further work section? It is not a result per se.

p11. Add a short discussion section first including further work.

p11. The conclusions are poorly written, not very readable and do not highlight the success of the work or results obtained. Please improve the whole section for further review. One example:

p11 line 15. "there were 112 recorded models used over a two-year period," 112 models? That is is not the case unless I have missed something.

p11 line 16 foward slash

p11 line 17 'outline data' I think you mean 'outlier data'

p11 lines 19 to 20 quantify etc.

---

## Referee Comment (RC2) · Anonymous Referee #2 · 2 Oct 2017

The paper deals with a study of modelling the optimum coagulant dose in jar test using the artificial neural network. Subject is important for the efficient water treatment. However, the manuscript needs major revisions before considered to be published. In general, please check the grammar and spelling, and remove the repetitions.

---

## Referee Comment (RC3) · Anonymous Referee #3 · 3 Oct 2017

In general, the paper was well prepared. Although it is understandable, the paper does not introduce a novel technique, however, this paper is still the need for some further improvements to meet this journal. Anyhow, the following comments should be considered carefully.

1- Grammatical and structural errors e.g., In the abstract, "flocculation in water treatment are among the common ways . . . "Should it be flocculation in water treatment are the common ways. . .? 2- Introduction is too long. 3- Data source should be mentioned and if the authors collaborated with field or laboratory tests or compiled them and also in which country. 4- This result (MSE and $R^2$ ) is based on normalized data, what happens if we use the real data for your new model? You should explain about this matter. 5- This paper proposes a new model for predicting Optimum Coagulant of

water treatment, but the discussion analysis of this new model is too few and simple, some related literature and discussion need to be added. 6- I feel the author could add more content in Section Results and Discussion. Now, this section is too simple, only is the introduction of the results. The findings need to be analyzed further in order to avoid the straightforward and nonsense.

Please also note the supplement to this comment:
https://www.drink-water-eng-sci-discuss.net/dwes-2017-24/dwes-2017-24-RC3-supplement.pdf

---

## Author Comment (AC1) · 30 Oct 2017

Dear Prof,

I greatly appreciate your consideration and time. I applied all the changes that you mentioned in your comments and thank you for introducing two new reference and I used them in my paper. Please let me know if you want me to review any other paper regarding my paper quality improvement.

regards, Amin

––––––––––––––––––––––––

---

## Author Comment (AC2) · 30 Oct 2017

Dear Prof,

I greatly appreciate your consideration and time. I applied all the changes that you mentioned in your comments. Please let me know if you want me to review any other paper regarding my paper quality improvement.

regards, Amin
* * *

---

## Author Comment (AC4) · 30 Oct 2017

Dear Prof,

Please find the attached file as my final version with and without track changes.

Regards, Amin

[revised manuscript text omitted]

---

## Author Comment (AC5) · 30 Oct 2017

Dear Prof,

Please find the attached file as my final version with and without track changes.

Regards, Amin
* * *
[Figure]

[revised manuscript text omitted]

---

## Author Comment (AC6) · 30 Oct 2017

Dear Prof,

Please find the attached file as my final version with and without track changes.

[revised manuscript text omitted]